# Ca^2+^ Signaling as the Untact Mode during Signaling in Metastatic Breast Cancer

**DOI:** 10.3390/cancers13061473

**Published:** 2021-03-23

**Authors:** Dongun Lee, Jeong Hee Hong

**Affiliations:** Department of Health Sciences and Technology, Lee Gil Ya Cancer and Diabetes Institute, GAIHST, Gachon University, 155 Getbeolro, Yeonsu-gu, Incheon 21999, Korea; sppotato1@gmail.com

**Keywords:** Ca^2+^ channels, breast cancer, Ca^2+^ signaling, brain metastasis

## Abstract

**Simple Summary:**

Intracellular Ca^2+^ signaling is a critical factor in breast cancer metastasis. In the proliferation stage, increases in intracellular Ca^2+^ concentration through voltage-dependent Ca^2+^ channels, P_2_Y_2_ channels, transient receptor potential (TRP) channels, store-operated Ca^2+^ channels (SOCCs), and IP_3_ receptors and a decrease in intracellular Ca^2+^ concentration through plasma membrane Ca^2+^ ATPases and secretory pathway Ca^2+^ ATPases (SPCA) activate breast cancer cell proliferation. TRPM7, SOCC, inositol trisphosphate receptor (IP_3_R), ryanodine receptor (RyR), and sarco-/endo-plasmic reticulum Ca^2+^-ATPase (SERCA) increase the expression of epithelial-to-mesenchymal transition (EMT)-related proteins; meanwhile, SPCA and the Na^+^/Ca^2+^ exchanger (NCX) control the activation of EMT-related proteins. Increased Ca^2+^ through TRPC1, TRPM7/8, P_2_X_7_, and SOCC enhances breast cancer cell migration. The stromal interaction molecule (STIM)-Orai complex, P_2_X_7_, and Ca^2+^ sensing receptors are involved in invadopodia. Various pharmacological agents for Ca^2+^ channels have been proposed against breast cancer and have provided potential strategies for treating metastatic processes.

**Abstract:**

Metastatic features of breast cancer in the brain are considered a common pathology in female patients with late-stage breast cancer. Ca^2+^ signaling and the overexpression pattern of Ca^2+^ channels have been regarded as oncogenic markers of breast cancer. In other words, breast tumor development can be mediated by inhibiting Ca^2+^ channels. Although the therapeutic potential of inhibiting Ca^2+^ channels against breast cancer has been demonstrated, the relationship between breast cancer metastasis and Ca^2+^ channels is not yet understood. Thus, we focused on the metastatic features of breast cancer and summarized the basic mechanisms of Ca^2+^-related proteins and channels during the stages of metastatic breast cancer by evaluating Ca^2+^ signaling. In particular, we highlighted the metastasis of breast tumors to the brain. Thus, modulating Ca^2+^ channels with Ca^2+^ channel inhibitors and combined applications will advance treatment strategies for breast cancer metastasis to the brain.

## 1. Introduction

Cancer metastasis occurs in several stages, including proliferation, epithelial-to-mesenchymal transition (EMT), invasion, transport, colonization, and angiogenesis (Figure 1) [1]. In fully developed tumorigenesis stages, circulating tumor cells move into another tissue and transform into mesenchymal stem cell-like cells as a result of EMT [2,3]. EMT is the initiation step in cancer metastasis [4]. Tumor cells are transported through the bloodstream after invading blood vessels [5,6,7] in a process called intravasation [8]. The metastasized tumor cells attach and grow via colonization; then, the blood vessels that supply nutrients are generated during angiogenesis, leading to cancer development [2,9,10]. In many stages of metastasis, the proteins and factors related to metastasis are intricate [11]. Therefore, messenger signaling to block metastasis and tumorigenesis is necessary for the fundamental processes that regulate initial signaling factors, but protein signaling is not. Breast cancer is the most common cancer type, and it has been considered one of the most malignant cancers in women worldwide [12,13]. Breast cancer subtypes include triple-negative and triple-positive breast cancer resulting from the existence and nonexistence of estrogen receptors, progesterone receptors, or human epidermal growth factor receptor-2 (HER2) [14,15]. Each subtype has the following cell lines: triple-negative (MDA-MB-231, MDA-MB-486, and MCF-10A [16,17]), triple-positive (BSMZ, BT474, and EFM192A [16]), and hormone receptor-positive cell lines that express estrogen receptors and progesterone receptors in the absence of HER2 (MCF-7 and T47D [16]). Genotypic or phenotypic heterogeneity of breast cancer is diverse. While triple-negative breast cancer generally has the most aggressive behavior and poor clinical outcomes [18,19,20], triple-positive breast cancer has also been found to exhibit aggressive behavior, despite the availability of antibody-targeted therapy or chemotherapy [21].

In this review, we elucidate the essential processes of metastasis in breast cancer. In particular, Ca^2+^ signaling molecules are introduced, and the processes involved in the fundamental modulation of Ca^2+^ signaling modules and potential strategies against breast cancer are addressed.

### 1.1. Ca^2+^ Signaling-Associated Molecules

The physiological role of Ca^2+^ signaling is commonly known to include muscle contraction to crosslink actin, myosin, and muscle fibers [22]. In addition, Ca^2+^ signaling regulates physiological and pathological cellular pathways, including cell proliferation, differentiation, migration, muscle contraction, neurotransmitter release, and fluid secretion [23,24,25]. The regulation of cellular Ca^2+^ as a key signaling messenger is precisely modulated by numerous Ca^2+^ channels and transporters associated with the membrane of intracellular compartments or plasma membranes [26]. The concentration of intracellular Ca^2+^ ([Ca^2+^]_i_) in the resting state is sustained up to 100 nM, allowing the use of evoked Ca^2+^ for the signaling pathways involved in cellular functions [26]. Generally, evoked intracellular Ca^2+^ passes through the endoplasmic reticulum (ER) membrane via two mechanisms: the outward movement to the cytoplasm through the inositol trisphosphate receptor (IP_3_R) [27,28,29] and the ryanodine receptor (RyR) from intracellular Ca^2+^ stores [30,31,32]. Elicited Ca^2+^ is attenuated by movement into the ER via sarco-/endoplasmic reticulum Ca^2+^-ATPase (SERCA) or movement to the extracellular matrix via plasma membrane Ca^2+^-ATPase (PMCA) [33,34]. The types of Ca^2+^ channels in the plasma membrane are discussed in the following section. RyR is activated by several molecules or drugs such as cADP ribose [35,36], 4-chloro-m-cresol [37], and suramin [38], and SERCA is stimulated by adenosine triphosphate (ATP) [34]. IP_3_R is activated by IP_3_ via extracellular signaling through the hydrolysis of phosphatidylinositol 4,5-bisphosphate by phospholipase C (PLC) [27,29,39].

### 1.2. Types of Ca^2+^ Channels

Plasma membrane-localized Ca^2+^ channels are classified into three types: voltage-gated Ca^2+^ channels (VGCCs), ligand-gated Ca^2+^ channels (LGCCs), and store-operated Ca^2+^ channels (SOCCs) [26]. VGCCs are stimulated by depolarization of the plasma membrane through a concentration gradient of [Ca^2+^]_i_ [40]. Exceptionally, the Na^+^/Ca^2+^ exchanger (NCX) is activated by changes in Ca^2+^ concentration to control [Ca^2+^]_i_ by exchanging Ca^2+^ and Na^+^ [41]. LGCCs in the plasma membrane are composed of numerous types of channels, such as ATP receptors and ionotropic glutamate receptors (e.g., α-amino-3-hydroxy-5-methyl-4-isoxazolepropionic acid receptors or *N*-methyl-d-aspartate receptors) [42,43,44]. LGCCs in the plasma membrane, which are structurally classified as G protein-coupled receptors (GPCRs), consist of seven transmembrane domains and have the largest number of subtypes in cell-surface receptor groups [45]. IP_3_R, RyR, and two-pore Ca^2+^ channels are also present in LGCCs. As there are numerous GPCR subfamilies, these receptors are targets of various therapeutic drugs [46]. In particular, GPCRs are stimulated by neurotransmitters and neurotransmitter-like agonists, including noradrenaline and cholinergic compounds and molecules such as carbamylcholine and vasopressin [47]. Stimulation of GPCRs is initiated by the phosphorylation of guanidine diphosphate for guanidine triphosphate through the α-subunit of G-proteins (Gα) [48]. Activated G-proteins dissociate Gα to deliver several intracellular signals according to each Gα subunit: Gs (adenylyl cyclase increases), Gi (adenylyl cyclase decreases), and Gq (PLC increases) [49]. Each signal mediates a tremendous number of cellular functions by increasing or decreasing Ca^2+^ signaling. Lastly, SOCCs are affected by decreasing the concentration of Ca^2+^ in the ER by delivering signals from the oligomerized stromal interaction molecule (STIM), which senses Ca^2+^ depletion on the ER membrane and subsequently activates Orai channels [50,51]. In addition, transient receptor potential (TRP) channels play various important roles in the cell life cycle, including the transduction of neurotransmitters and immunization, and are activated by temperature, pH, and specific compounds [52,53]. Furthermore, SOCC-related proteins, including Orais and STIMs, show different characteristics against the estrogen receptor [54]. In the estrogen receptor-positive cell line MCF-7, store-operated Ca^2+^ entry (SOCE) is mediated by the combination of STIM1/2 and Orai3; meanwhile, in the estrogen receptor-negative cell line MDA-MB-231, SOCE is mediated by the combination of STIM1 and Orai1 in activated SOCCs [54].

Recent studies have demonstrated that Ca^2+^ plays a crucial role not only in malignant proliferation but also in cancer metastasis [55,56,57,58,59,60]. Breast cancer is considered a metastatic cancer due to its aggressive features [61]. Thus, in this review, we focus on the characteristics of breast cancer in the context of Ca^2+^ signaling and cancer metastasis, especially from breast to brain. Furthermore, we discuss potential strategies to overcome the disadvantages of breast cancer-targeted therapy, taking Ca^2+^ signaling into consideration.

## 2. The Relationship between Breast Cancer Metastasis and Ca^2+^ Channels

The large range of physiological, pharmacological, and clinical roles of Ca^2+^ signaling in breast cancer are well known, including the expression patterns of the channels, the effect of channel activity on tumorigenesis, and therapeutic targets [62,63,64]. Several Ca^2+^-related proteins and channels are overexpressed in breast cancer cells, including IP_3_R [65], Orais [66,67,68], PMCA [69,70], and TRP channels [71,72,73,74]. Each channel plays a critical role in cancer cell viability. In this section, we summarize the effects of Ca^2+^ channels on the growth and metastatic stages of breast cancer.

### 2.1. Proliferation in the Initial Metastatic Stage

Cancer metastasis is initiated from an excessively developed primary tumor and its subsequent transport to the bloodstream [1]. To treat metastatic tumors, their vigorous proliferation and immoderately active cell cycle must be controlled to block cancer growth. One of the Ca^2+^-ATPases, secretory pathway Ca^2+^-ATPase (SPCA, localized on Golgi and transports Ca^2+^ into the Golgi), has been shown to be activated in breast cancer. This pathway induces tumorigenesis by activating extracellular signal-regulated kinase (ERK)1/2 activity and increasing tumor proliferation [75,76]. SPCA1-silenced MDA-MB-231 (triple-negative) cells with SPCA1 siRNA exhibit a lower rate of cell growth and decreased insulin-like growth factor receptor expression [75]. SPCA2 shows different expression patterns according to the presence of the estrogen receptor [76]. SPCA2 is overexpressed in estrogen receptor-positive MCF-7 cells, but is barely expressed in estrogen receptor-negative MCF-10A cells [76]. SPCA2 silencing with shRNA decreases MCF-7 cell proliferation; however, SPCA2 overexpression increases MCF-10A cell proliferation [76]. SPCA2 can induce store-independent Orai1 Ca^2+^ influx [77]. The increased proliferation of breast cancer cells (MCF-7 wild-type and SPCA2-overexpressed MCF-10A) is induced by Ca^2+^ influx from SPCA2-stimulated Orai1 [76]. Inhibition of PMCA2 decreases cancer proliferation [78] and leads to cancer cell death [79] in MDA-MB-231 cells. Additionally, downregulation of PMCA2, which is overexpressed in MDA-MB-231 cells, enhances the anticancer effect of doxorubicin [78]. T-type VGCC, Ca_V_3.1, and Ca_V_3.2 are blocked by NNC-55-0396, a T-type Ca^2+^ channel blocker, and each relevant siRNA attenuates breast cancer proliferation regardless of estrogen receptor-positive or -negative cell lines (MCF-7, MDA-MB-231, and MCF-10A) [80]. Increased [Ca^2+^]_i_ through IP_3_R3 induces MCF-7 breast cancer cell growth [81]. Additionally, TNF-αinduces the release of ATP, and the ATP-stimulated Ca^2+^ channel P_2_Y_2_ receptor induces tumor growth and invasion of MDA-MB-231 cells (estrogen receptor-negative), but MCF-7 (estrogen receptor-positive) cells, which has a low metastatic feature, induces less release of ATP and reveals low P_2_Y_2_ receptor activation [82].

TRP channels play critical roles in cell viability by increasing channel activities, including TRP canonical (C), TRP melastatin (M), and TRP vanilloid (V) in breast cancer cells and tissues [74,83,84,85]. TRP channels are classified into six subfamilies, including TRPC, TRPM, TRPV, TRP ankyrin (A), TRP canonical (C), and TRP mucolipin (ML), which are composed of six transmembrane domains [52,53]. These nonselective Ca^2+^-permeable channels have a superfamily of at least 20 subtypes that function in various ways [26,52] and can regulate breast cancer tumorigenesis. TRPC channels, including TRPC3, TRPC5, and TRPC6, are typical oncogenic proteins that can be used to diagnose breast cancer [72,84,86]. Increases in extracellular Ca^2+^ concentration ([Ca^2+^]_ex_) induce overexpression of the TRPC1 channel and increase proliferation through epidermal growth factor receptor (EGFR) signals with ERK1/2 phosphorylation in MCF-7 cells [83]. TRPC6 is more highly expressed than other TRPC channels in human breast cancer MCF-7/MDA-MB-231 cell lines (regardless of breast cancer subtypes) and tissues, but TRPC3 is highly expressed only in the estrogen receptor-negative MDA-MB-231 cells and tissues [84]. In particular, activated TRPC6 increases cellular proliferation [84]. TRPM7 is overexpressed in human breast adenocarcinoma tissue, whereas TRPM7 silencing attenuates MCF-7 cell proliferation [85]. Increased [Ca^2+^]_i_ induces cell proliferation through TRPV6-mediated Ca^2+^ influx [87] and TRPV6 inhibition, which are upregulated by sex hormones leading to T47D (hormone receptor-positive) cancer cell death [74]. A schematic illustration of Ca^2+^ channels that increase breast cancer proliferation is shown in Figure 2.

### 2.2. EMT

To invade blood vessels, primary tissue cells must undergo cellular shape and construction transformation, which is known as EMT. In other words, epithelial tumor cells that reside in the primary tumor tissue turn into mesenchymal tumor cells that surround the bloodstream and invade the extracellular matrix (ECM) [3,88]. During EMT, the expression of several Ca^2+^ channels and transporters is modulated by transforming growth factor (TGF-β) [89] and epidermal growth factor (EGF) [90] in human breast cancer cells (Figure 3). In TGF-β-induced MCF-7 EMT, IP_3_R, and SERCA3 proteins are overexpressed, while NCX1 is downregulated [89]. TGF-β stimulates store-operated Ca^2+^ entry (SOCE) by upregulating STIM1 and Orai1 expression through the overexpression of the transcription factor Oct4 in MCF-7 cells (estrogen receptor-positive), but not in MDA-MB-231 cells (estrogen receptor-negative) [91]. SERCA2, IP_3_R1/3, RyR2, and Orai1 are overexpressed during EGF-induced EMT in MDA-MB-468 [90,92]. Additionally, EGF-stimulated EMT, which transforms breast cancer MDA-MB-468 cells into a mesenchymal-like shape, increases the expression of ATP-binding cassette subfamily C member 3 (ABCC3) after Ca^2+^ signaling of TRPC1 [93].

The Ca^2+^-ATPase SPCA2, which is encoded by the *ATP2C2* gene, is an epithelial marker that inhibits cell-adhesion protein E-cadherin biogenesis in breast cancer cells regardless of estrogen receptor existence (MCF-7 and MDA-MB-231) [94]. Activation of SPCA induces cell-to-cell contacts and continuously stimulates E-cadherin-induced Hippo-YAP signaling to inhibit EMT formation in MCF-7 cells [94]. Meanwhile, SPCA2-silenced MCF-7 cells show EGF-induced expression of EMT-related proteins, including zinc finger E-box-binding homeobox 1, N-cadherin, snail family transcription repressor 2, fibronectin, and vimentin [94]. Interestingly, SPCA2 upregulation in MDA-MB-231 cells decreases EMT-related protein expression and attenuates metastasis in the MDA-MB-231-injected breast cancer mouse model [94]. SOCE intensifies TGF-β-induced EMT in MDA-MB-231 and MCF-10A cells by upregulating STIM1 [95] and TRPC1 [96]. Additionally, TRPC1 is involved in the regulation of hypoxia-induced EMT in MDA-MB-468 cells [97]. As EMT regulatory factors, signal transducer and activator of transcription 3 (STAT3) phosphorylation and vimentin expression are induced by Ca^2+^ signaling through TRPM7 [98]. To initiate EMT, the binding among primary cells must collapse, and there must also be a reduction in the expression of the binding protein E-cadherin [99]. Thus, Ca^2+^ signaling can attenuate E-cadherin expression [100]. EMT is also activated by hypoxia and vascular endothelial growth factor (VEGF) [101]. Furthermore, ATP-mediated Ca^2+^ increase-stimulated EMT is induced by the stimulation of EGF and hypoxia in MDA-MB-468 and MDA-MB-231 cells [102,103]. Treatment with a Ca^2+^ chelating agent such as 1,2-bis (*o*-aminophenoxy)ethane-*N*,*N*,*N′*,*N′*-tetraacetic acid (BAPTA) or ethylene glycol-bis (β-aminoethyl ether)-*N*,*N*,*N′N′*-tetraacetic acid (EGTA) also attenuates EGF- or hypoxia-induced EMT, which is stimulated by TRPM7 Ca^2+^ signaling in MDA-MB-468 cells [98].

### 2.3. Migration and Intravasation

Breast cancer cells are transported to target tissues by migrating through blood vessels and invading the bloodstream. Intracellular Ca^2+^ signaling regulates cellular movement with proteins that induce migration (Figure 4), including myosin light chain kinase [104], myosin II [105,106], calpain [107], Ca^2+^/calmodulin-dependent protein kinase II (CaMKII) [108,109], and focal adhesion kinase (FAK) [110]. In breast cancer cells (MDA-MB-231), SOCE induces cellular migration [111]. Knockdown of Orai1 mRNA expression attenuates breast cancer cell migration, and Orai1 overexpression induces migration through increased Ras and Rac levels, using defects in focal adhesion to move the cells [111]. Migratory cells are needed to alter cytoskeletal structures with the formation of invadopodia to adhere to the forward region in the direction of progress and degrade the ECM [112,113]. STIM1 knockdown in MDA-MB-231 cells attenuates invadopodia formation and cancer invasion [114]. Additionally, elevated [Ca^2+^]_i_, which is induced by phospholipase C, activates the ERK 1/2 signaling pathway and subsequently stimulates MDA-MB-231 breast cancer cell migration [115]. The expression of each TRP subtype can be distinguished according to the histologic grade and invasive degree of cancer [72]. TRPM8 and TRPC1 are generally expressed in tissues smaller than 2 cm and in grade I tissue, and TRPM7 is expressed in tissue larger than 2 cm and in grade II tissue [72]. In addition, TRPM7 induces breast cancer migration regardless of the cellular subtypes (MDA-MB-231 and MCF-7) [116]. Although TRPC6 and TRPV6 have no relationship with histologic grade, TRPV6 is found in the invasive area, and silencing of TRPV6 expression reduces breast cancer cell migration [72].

For intravasation, invadopodia must be established [112]. Invadopodia formation is initiated by the activation of the nonreceptor tyrosine kinase Src (proto-oncogene tyrosine-protein kinase) [117,118,119]. STIM1-Orai1 stimulation induces Src activation and recruits metalloproteinases that degrade the ECM, leading to invadopodia formation in MCF-7 cells [120]. The ATP-gated channel P_2_X_7_ induces the release of gelatinolytic cysteine cathepsins from invadopodia to degrade ECM in MDA-MB-435 cells (triple-negative breast cancer) [121]. [Ca^2+^]_ex_ and [Ca^2+^]_i_ also regulate the intravasation of breast cancer cells through the extracellular Ca^2+^-sensing receptor (CaSR) with mitogen-activated protein kinases [122]. Increased [Ca^2+^]_ex_ activates CaSR and subsequently stimulates EGFR to induce invasion, whereas CaSR knockdown attenuates the intravasation of MDA-MB-231 cells [122].

### 2.4. Colonization and Angiogenesis

Tumor cells invading other tissues colonize by clustering their cells to form tumor tissue and carry out angiogenesis, thus creating an appropriate environment for tumorigenesis. Although it has a significant role in brain metastasis, the relationship between colonization and Ca^2+^ signaling has not been clearly elucidated. The mitochondrial Ca^2+^ uniporter is located at the mitochondrial membrane of breast cancer cells and regulates tumor proliferation [123,124,125], and triple-negative breast cancer cells (MDA-MB-231 and BT-549) express this transporter to a greater degree than non-triple-negative breast cancer cells (T47D, BT-474, and MCF-7) [126]. Downregulation of the mitochondrial Ca^2+^ uniporter through extracellular vesicles suppresses MDA-MB-231 cell colonization [126]. Additionally, the Ca^2+^-binding protein S100A4 promotes MDA-MB-231 cell colonization [127,128]. S100A4 is an agonist of GPCR signaling, including the PLCβ-IP_3_ pathway [129], which is known to promote metastasis in breast cancer [130]. S100A4 knockdown reduces the brain colonization of MDA-MB-231 cells [127], and S100A4 expression is increased in colonized MDA-MB-231 cells [128].

IP_3_R is located within the ER membrane and is also expressed in the nuclear envelope [131]. Ca^2+^ signaling through IP_3_R in the nucleus plays a critical role in inducing angiogenesis in breast cancer cells (MDA-MB-468) and regulates angiogenesis-related genes, including early growth response-1, C-X-C motif chemokine ligand 10 (CXCL10), C-C motif chemokine ligand (CCL)-2, and dentin matrix acidic phosphoprotein 1 (DMP1) [132]. S100A4 plays critical roles in various angiogenic pathways in MDA-MB-231 cells through upregulation of matrix metalloproteinase-13 [133], TGF-β1-induced ERK1/2 signaling [134], and osterix, which is a transcription factor for bone formation [135]. In addition, a recent study showed that SOCE activity is elevated by angiotensin-converting enzyme (ACE)2/angiotensin-1(1–7) according to breast cancer cell subtypes [136]. (ACE)2/angiotensin-1(1–7), which is more highly expressed in MCF-7 cells than in MDA-MB-231 cells, increases SOCE to inhibit migration [136]. (ACE)2/angiotensin-1(1–7)-silenced MCF-7 cells show decreased migration, while (ACE)2/angiotensin-1(1–7)-overexpressing MDA-MB-231 cells show increased migration [136].

## 3. Metastasis of Breast Tumor to the Brain

Brain metastasis from breast cancer is the most common, and secondary brain tumors from breast cancer are diagnosed more often than primary brain tumors [137]. Metastatic features of cancer cells toward specific organs were demonstrated in Stephen Paget’s seed and soil theory, which was analyzed using autopsy records of breast cancer patients [138]. The microenvironment of cancer cells facilitates metastasis (seed growth) under favorable circumstances. Paget postulated that various cytokines secreted from cancer cells, including CCL2, CCL5, and interleukin-6, and a specific microenvironment communicates to attract cancer cells toward specific organs [138]. Accordingly, roughly 20% of breast cancer patients show central nervous system metastases, and these cases are increasing [139].

Ca^2+^ signaling in breast cancer is prominent in breast cancer metastasis. Therefore, controlling the activity of Ca^2+^ channels in breast tumor cells can lead to new therapeutic methods for brain metastases resulting from breast cancer. Several studies have suggested Ca^2+^ channels as new therapeutic targets for metastasis. Additional investigations have addressed the attenuation of Ca^2+^ signaling, which modulates the adhesive function and permeability. For example, MDA-MB-231 cells can cross the human brain microvascular endothelial cell (HBMEC) monolayer by stimulating vascular permeability factor (VPF) [140] and stromal cell-derived factor-1α (SDF-1α) [141]. VEGF/VPF stimulation also increases MDA-MB-231 cell adhesion onto the HBMEC monolayer and induces the redistribution of F-actin and disruption of vascular endothelial cadherin, which increases migration [140]. Treatment with the Ca^2+^-chelating agent BAPTA attenuates VEGF/VPF-induced cell adhesion, F-actin redistribution, and cadherin disruption [140]. Treatment with SDF-1α increases FAK phosphorylation by stimulating PI-3K signaling, which increases MDA-MB-231 cell migration, and SDF-1α is overexpressed in breast tissues compared to in normal tissues [141]. BAPTA attenuates SDF-1α-induced cellular permeability in a co-culture of MDA-MB-231 cells and HBMECs [141]. Beyond the role of Ca^2+^ signaling in adhesion and permeability, additional mechanisms of cell-cell crosstalk present a challenge in the metastatic process.

Additionally, Sharma et al. showed that regulating Ca_V_3.2 through specific radiofrequencies with an amplitude of 27.12 MHz attenuated brain metastatic breast cancer cells in vivo [142]. The study demonstrated that specific frequencies modulate several cancers in patients receiving noninvasive cancer therapy. The mouths of the patients were used to deliver frequencies via antenna, and it was found that a frequency of 27.12 MHz was breast cancer-specific [142]. These frequencies revealed antitumor effects in a xenograft mouse model and in brain tumor patients, suppressing brain metastases from breast cancer [142]. As previously mentioned, activation of Ca_V_3.2 through 27.12 MHz frequencies increases Ca^2+^ influx to the activated p38 pathway, which attenuates tumor progression [142].

Mechanosensitive Ca^2+^ channels are also involved in the metastasis of cells. Piezo channels, expressed in MCF-7 cells, regulate intracellular functions such as integrin activity in HeLa cells [143], regulation of neuronal-glial specification in human neuronal stem cells [144], maintenance of homeostatic cell numbers in epithelia [145], and sensing confinement of Chinese hamster ovary (CHO) cells [146]. Piezo2 is involved in mechanotransduction and force transmission in MDA-MB-231 cells [147]. Piezo2 activation is required for actin cytoskeletal reorganization and FAK phosphorylation through Fyn kinase [147]. Piezo2 activates the RhoA signaling cascade to promote brain metastasis in breast cancer. Piezo2 knockdown decreases the invasion of MDA-MB-231-BrM2 cells, which metastasize cells from breast cancer to the brain [147]. Additionally, when triple-negative breast cancer cells migrate to the brain, astrocytes activate the S100A4-related pathway (protocadherin 7 (PCDH7)-PLCβ-Ca^2+^-CaMKII/S100A4) [148]. In brain metastasis tissues from patients, PCDH7 expression is higher than that in lung metastasis tissue and mediates cellular interaction between astrocytes and cancer cells [148]. Furthermore, PCDH7 expression induces the penetration of tumor cells over the blood-brain barrier, which then increases tumor cell intravasation in the brain [148]. The cell-to-cell interaction between mouse astrocytes and MDA-MB-231 cells in the PCDH7-stimulated mouse model activated PLCβ-Ca^2+^-CaMKII/S100A4 signaling in MDA-MB-231 cells [148]. Moreover, the brain is highly responsive to estrogen [149], and brain metastasis is revealed in estrogen receptor-positive areas. Interestingly, estrogen receptor-negative breast cancer cells can be affected by estrogen through the involvement of astrocytes [127]. When astrocytes are stimulated by estrogen, the migratory ability of MDA-MB-231 cells cocultured with astrocytes reportedly increases, according to a wound-healing migration assay [127]. In this case, silencing S100A4 expression attenuates astrocyte-induced migration and colonization of MDA-MB-231 cells [127]. As mentioned previously, Ca^2+^ signaling is crucial for cancer development and progression. Therefore, more studies focusing on the relationship between cancer and Ca^2+^ should be conducted.

## 4. The Pharmacological Application of Ca^2+^ Signaling Blockers to Breast Cancer

Various attempts to use antagonists of Ca^2+^ channels have been proposed to control breast cancer tumorigenesis. The mediation of [Ca^2+^]_i_ signaling is critical for cellular functions regardless of the cellular type (tumor vs. nontumor). In other words, the application of Ca^2+^ signaling blockers for anticancer drugs requires in-depth studies of the basic mechanisms underlying Ca^2+^ signaling and cancer cells. Thus, we summarized the studies that have used Ca^2+^ channel blockers for breast cancer medication to understand the associated mechanisms (Table 1). Recent studies have shown that the L-type Ca^2+^ channel blockers amlodipine, diltiazem, and verapamil have been used to modulate high blood pressure [150,151,152] and attenuate HT39-transplanted breast cancer growth [153]. Mice with increased Ca^2+^ concentration in serum exhibit a larger amount of HT39 tumor tissue, while treatment with amlodipine attenuates Ca^2+^ signaling in HT39 cells with a decrease in tumor size [153]. The T-type Ca^2+^ channel blockers mibefradil (another hypertension drug [154]) and pimozide (chronic psychosis drug [155]) inhibit MCF-7 breast cancer cell growth by inhibiting T-type Ca^2+^ current; furthermore, combined treatment with pimozide and mibefradil shows synergistic effects on cell growth in MCF-7 cells, decreasing cell growth [156].

As mentioned in Section 2.1, TRP channels are prominent in breast cancer. Among these, TRPM channels are considered therapeutic targets for antagonists. The TRPM7 inhibitor 2-aminoethyl diphenylborinate (2-APB [157]) attenuates MDA-MB-231, AU565, and T47D cell proliferation, increasing S phase and decreasing G0/G1 phase in the breast cancer cell cycle [158]. Moreover, TRPM7-silenced MDA-MB-231 cells have no antitumor effects when 2-APB is administered [158]. Treatment with the antifungal agent clotrimazole, which inhibits TRPM2 activity [159], decreases MDA-MB-231 cell invasion, which is accompanied by apoptosis and G1-phase arrest [160]. Clotrimazole increases cleaved poly (ADP-ribose) polymerase (PARP), cleaved caspase-3, and B-cell lymphoma-2 (Bcl-2)-associated X expression, which induces apoptotic signaling in MDA-MB-231 cells [160]. Inhibition of Ca^2+^ signaling with the voltage-independent Ca^2+^ channel inhibitor carboxyamidotriazole reduces MCF-7 proliferation by arresting G2/M phase cell cycle, decreasing BCL-2 (which blocks apoptotic signaling) expression, and increasing p21 expression, which induces apoptotic signaling [161]. Furthermore, treatment with carboxyamidotriazole reduces mitochondrial membrane potential [161], which is highly activated in cancer stem cells to produce reactive oxygen species (ROS) [162]. In addition, administration of the SERCA inhibitor thapsigargin inhibits S100A4 protein expression in MDA-MB-231 breast cancer cells [163].

**Table 1 cancers-13-01473-t001:** The Ca^2+^ channel blockers with potential anticancer effects.

Reagents	Description	Effect	Ref.
Amlodipine	Medication for high blood pressure and L-type Ca^2+^ channel inhibitor	Decrease of HT39-transplanted breast cancer growth	[153]
Diltiazem			
Verapamil			
Mibefradil	Hypertension drug	Decrease of MCF-7 growth through inhibition of T-type Ca^2+^ current	[154]
Pimozide	Chronic psychosis drug		[155]
2-APB	TRPM7 inhibitor	Decrease of MDA-MB-231, AU565, and T47D cell growth through pausing cell cycle	[158]
Clotrimazole	TRPM2 inhibitor	Decrease of MDA-MB-231 cell growth through G1-phase arrest	[160]
Carboxyamidotriazole	Reduce mitochondrial membrane potential	Attenuation of ROS	[162]
Thapsigargin	SERCA inhibitor	Inhibition of S100A4 expression in MDA-MB-231	[163]

In addition, Ca^2+^ channel blockers enhance the therapeutic effect of traditional drugs or overcome resistance to insignificant drugs. In an attempt to improve their therapeutic effect on breast cancer, mibefradil enhanced the apoptotic effect of the anticancer drug 2-deoxy-D-glucose (2-DG) by arresting the cell cycle in MDA-MB-231 cells [164]. Furthermore, clotrimazole increases the inhibitory effect of imatinib mesylate on T74D cells to mediate kinase inhibition [165]. Mibefradil is a T-type Ca^2+^ channel blocker that arrests the cell cycle at the G1 phase and evaluates glucose metabolism [164]. The application of only 2-DG also inhibits MDA-MB-231 cell growth. Although the inhibition rate is very low (approximately 10%), the combination of mibefradil and 2-DG leads to a synergistic antitumor effect (approximately 30% of inhibition rate) [164]. The combination of imatinib mesylate and clotrimazole synergistically decreases T74D cell growth by increasing lactate dehydrogenase and nitric oxide leakage [164], which induces membrane damage and apoptosis in cancer cells [166,167]. Doxorubicin and daunorubicin are the most well-known anthracycline antibiotics and are also first-line drugs for malignancies [168]. They have structural features that can be intercalated into DNA bases and inhibit topo ii/DNA ternary complexes [169]. Additionally, the quinone ring, a common structure for anthracyclines such as doxorubicin and daunorubicin, induces ROS production [170,171,172,173]. Doxorubicin and daunorubicin are typical anticancer reagents; however, they are hindered by multidrug resistance in breast cancer [174,175]. The addition of diltiazem to doxorubicin-treated MCF-7 cells increases the expression of apoptosis-related p53 genes [174]. The combination of daunorubicin and amlodipine reportedly predominantly attenuates tumor volume in the MCF-7 xenograft tumor model via mitochondrial destruction [175]. Despite these applicable combinations, more studies on effective combinations of Ca^2+^ channel blockers and traditional anticancer drugs should be conducted. These combined treatments are suggested as novel therapeutic strategies against breast cancer and breast-to-brain metastatic cancer.

As mentioned above, Ca^2+^ channel blockers have pharmacological potential. However, the therapeutic application of Ca^2+^ channel blockers is challenging, as each reagent does not act on a single channel or transporter. The TRPM7 inhibitor 2-APB inhibits IP_3_R [176], Orai1/2-induced SOCE [177], and other TRP channels [178]. In contrast, 2-APB induces Orai3-induced Ca^2+^ influx [177]. Additionally, the Ca^2+^ channel blocker clotrimazole can inhibit Ca^2+^-activated potassium channel 3.1 [179], which drives Ca^2+^ through SOCE [180]. Although several Ca^2+^ channel blockers are pharmacologically complicated to use as therapeutic strategies, the specific mechanisms of Ca^2+^ channel blockers need to be clarified.

## 5. Future Perspective

The relationship between Ca^2+^ channels and breast cancer has been assessed for several decades; however, the effect of Ca^2+^ channels on the metastasis of breast cancer to the brain requires further investigation. The treatment of breast cancer by modulating Ca^2+^ channel expression and its activity has been considered a cancer therapeutic strategy using various Ca^2+^ channel blockers. Although Ca^2+^ signaling is closely related to cancer metastasis in various organs, the application of Ca^2+^ channel modulation for breast cancer metastasis has not been sufficiently studied. Based on the scope of metastatic breast cancer in this review, several studies have shown that Ca^2+^ channels have the potential to control metastatic stages and the movement of metastatic breast cancer cells to the brain by modulating adhesive function and permeability. Over the past several years, the number of cases of brain metastases from breast cancer has increased, and the entire metastatic process has not been fully elucidated. In addition, other metastatic processes should be highlighted beyond adhesive and invasive processes. For example, cellular-secreted processes and gene transcription activities are associated with Ca^2+^ signaling. In other words, communication between cancer cells and other tissues will commence with the untact mode, such as cytokine release. This mode builds up prior to the contact mode, which includes adhesion. As mentioned at the beginning of this article, Ca^2+^ is an attractive source of the untact mode for transferring the on-mode of metastatic signals through simple mobilization from abundant sources. Therefore, blocking Ca^2+^ channels as gatekeepers and modulating Ca^2+^ signaling can be attractive candidates for therapeutic approaches, and suitable combination therapies are suggested as relevant options for metastatic breast cancer therapy.

## Figures and Tables

**Figure 1 cancers-13-01473-f001:**
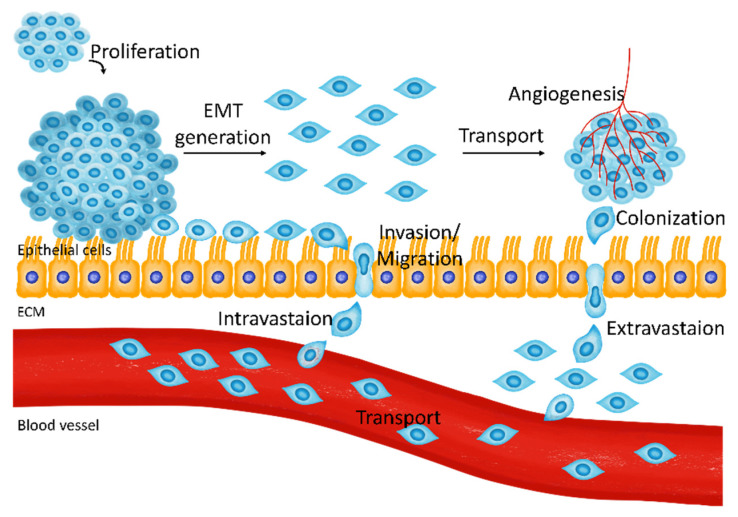
The metastatic pathway of breast cancer cells. Proliferated breast cancer cells are transformed into mesenchymal-like cells and undergo invasion and intravasation to blood vessels. Transporting tumor cells perform extravasation from blood vessels and generate a cancerous environment through colonization and angiogenesis.

**Figure 2 cancers-13-01473-f002:**
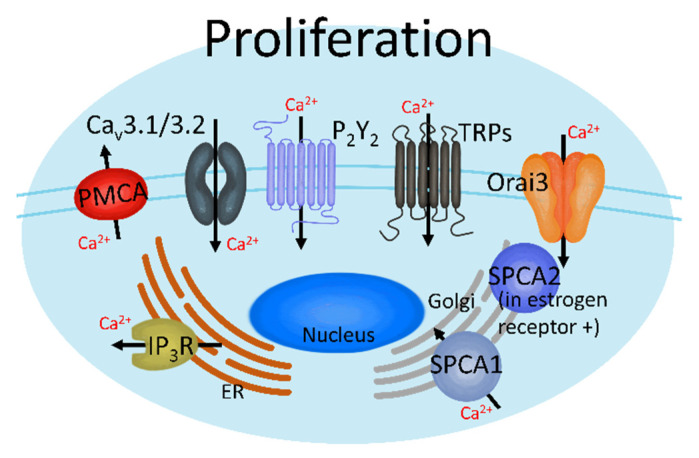
Ca^2+^ channels induce breast cancer proliferation. These channels can regulate breast cancer cell proliferation by activating each Ca^2+^ transporter, which increases or decreases intracellular Ca^2+^ ([Ca^2+^]_i_).

**Figure 3 cancers-13-01473-f003:**
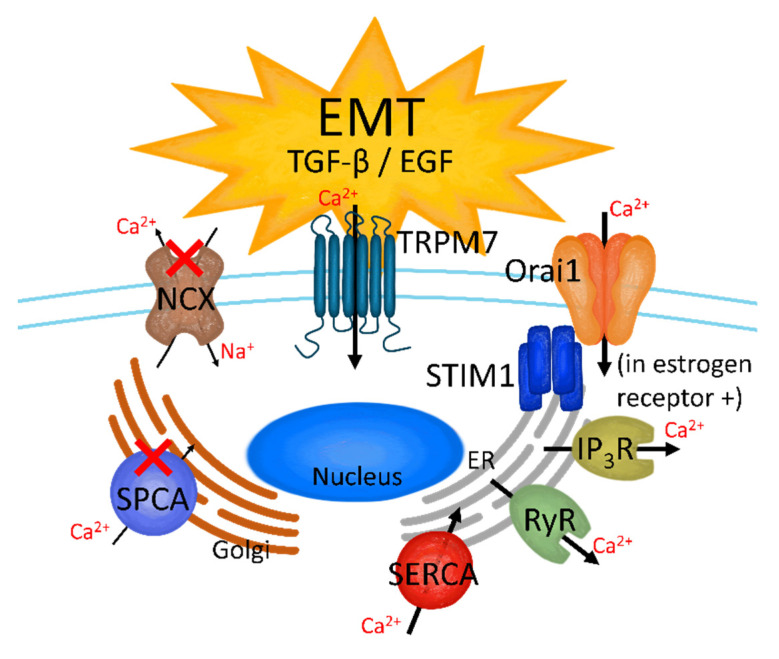
The effect on Ca^2+^ channels in TGF-β- and EGF-induced EMT breast cancer cells. TRM7, store-operated Ca^2+^ channels (SOCC), inositol trisphosphate receptor (IP_3_R), ryanodine receptor (RyR), and sarco-/endo-plasmic reticulum Ca^2+^-ATPase (SERCA) are activated when the breast cancer cells are stimulated by TGF-β or EGF; on the other hand, SPCA and NCX are downregulated in the TGF-β- and EGF-induced EMT stage.

**Figure 4 cancers-13-01473-f004:**
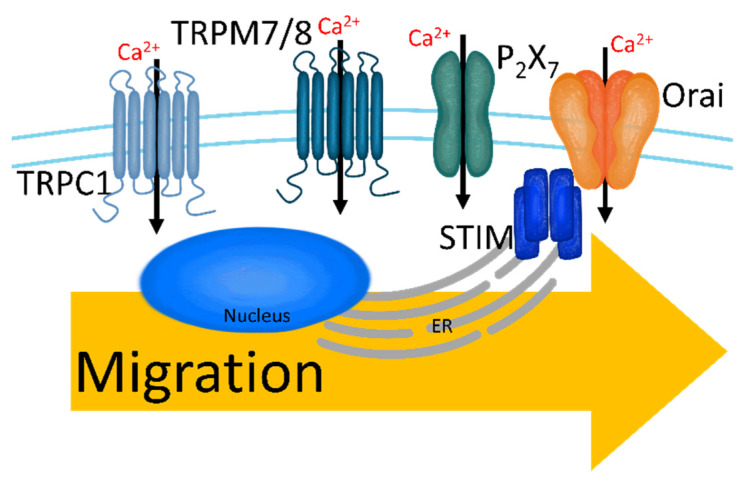
The upregulated Ca^2+^ channels in migratory breast cancer cells. Increased [Ca^2+^]_i_ through TRPC1, TRPM7/8, P_2_X_7_, and SOCC induces migration and intravasation.

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
