# Peer review of "Ca2+ Signaling as the Untact Mode during Signaling in Metastatic Breast Cancer"

_cancers, 2021, doi:10.3390/cancers13061473_

Round 1
Reviewer 1 Report
Comments on Cancers 1129639
This MS is a review about the role of Ca2+ signaling in metastatic breast cancer.
Despite a scientific interest, this MS suffers from simplification. One is that people use to talk about breast cancer, but in really it is breast cancers. And from the literature, the Ca2+ signaling in the different kind of BCs is different and implies different types of Ca2+ transporters. For example, it was shown 10 years ago, that ER+ samples and cell lines (MCF7) arbored an Orai3-dependent Ca2+ entry, when ER- samples and cell lines (MDA-MB231) have an Orai1-dependent Ca2+ entry. Furthermore, the role of Orai1 and Orai3 are not really analysed for the cell physiology and metastasis.
It is really dangerous to try to do a review with so different cell lines, with different behaviours.
Pharmacology is also an interesting field. But a dangerous one. For example, the authors cited an article about the use of 2-APB on MDA-MB231 cell line to inhibit TRPM7. EC50 is between 70-170 µM. But MDA-MB231 cells express an Orai1-dependent SOCE that can be fully inhibited by 50 µM 2-APB… Clotrimazole is also known as an inhibitor of some Ca2+-activated K+ channels, which increase the Ca2+ driving force… Noteworthy, 30 µM 2-APB is an opener of Orai3 channels, meaning that 2-APB effect on MCF-7 cell is totally different, and conclusions about 2-APB therapeutic potential on BC are tricky.
I recommend the authors to reorganize their MS by focussing on one by one BC type, and the associated cell line: example of triple-negative receptor BC + MDA-MB231 cell line / triple positive receptor BC + MCF7 cell line.
Furthermore, they should focus a bit on Orai channels and their relationship with SPCA.
Author Response
Dear reviewer and editor,
Before addressing each of the comments below, we appreciate the reviewers for the valuable comments and careful consideration. We obviously have needed to quote all sources correctly and done so at the places where we had missed before. In addition, the manuscript has been edited to make appropriate information and additional data to this body of work.
Responses to comments of reviewer as below:
Reviewer 1
This MS is a review about the role of Ca2+ signaling in metastatic breast cancer.
Despite a scientific interest, this MS suffers from simplification. One is that people use to talk about breast cancer, but in really it is breast cancers.
And from the literature, the Ca2+ signaling in the different kind of BCs is different and implies different types of Ca2+ transporters. For example, it was shown 10 years ago, that ER+ samples and cell lines (MCF7) arbored an Orai3-dependent Ca2+ entry, when ER- samples and cell lines (MDA-MB231) have an Orai1-dependent Ca2+ entry. Furthermore, the role of Orai1 and Orai3 are not really analysed for the cell physiology and metastasis. It is really dangerous to try to do a review with so different cell lines, with different behaviours.
Pharmacology is also an interesting field. But a dangerous one. For example, the authors cited an article about the use of 2-APB on MDA-MB231 cell line to inhibit TRPM7. EC50 is between 70-170 µM. But MDA-MB231 cells express an Orai1-dependent SOCE that can be fully inhibited by 50 µM 2-APB… Clotrimazole is also known as an inhibitor of some Ca2+-activated K+ channels, which increase the Ca2+ driving force… Noteworthy, 30 µM 2-APB is an opener of Orai3 channels, meaning that 2-APB effect on MCF-7 cell is totally different, and conclusions about 2-APB therapeutic potential on BC are tricky.
-Response: We appreciate your valuable comment and we agreed your comment. We’d like to summarize the role of calcium signaling modulator and removed overwhelming sentences. We added this point as you recommended (line 397).
I recommend the authors to reorganize their MS by focussing on one by one BC type, and the associated cell line: example of triple-negative receptor BC + MDA-MB231 cell line / triple positive receptor BC + MCF7 cell line.
-Response: We appreciate your valuable comment and we represented the type of breast cancer cell lines (line 46) and represented the type in the whole text.
Furthermore, they should focus a bit on Orai channels and their relationship with SPCA.
-Response: We appreciate your valuable comment and we added the relationship between SPCA and Orai (line 134).
Reviewer 2 Report
This review paper summarizes the basic mechanisms of calcium selective channels implicated in breast cancer and their role in metastatic stages of this malignance.
The topic touched upon in the article is relevant. The scientific content of the manuscript could justify its publication, but this manuscript requires thorough revision with extensive rewriting (possibly availing to the English editing service). This paper would benefit from some closer proofreading. It includes many linguistic errors (e.g. agreement of verbs) that at times make it difficult to follow. It may be useful to engage a professional English language editor following a restructure of the paper.
For instance:
Simple summary: ….”blockers are proposed against breast cancers and provide”….should be ….”blockers have been proposed against breast cancers and provide”
Abstract: The last sentences starting with “Thus, we demonstrated….are hard to follow. Please rewrite it in a clearer way.
Introduction, line 4: “that is transformed into a mesenchymal cell-like morphology owing to EMT”. This sentence is hard to follow. Please rephrase.
Introduction: “In this review, we elucidate the essential process of metastasis in breast cancer. In particular, Ca2+ signaling molecules are introduced and Ca2+ signaling-associated breast cancer metastasis is focused upon. This review also describes the fundamental modulation of Ca2+ signaling modules and discusses potential strategies against breast cancer”. Please rewrite these sentences-
Chapter 1.1: “The physiological role of Ca2+ signaling is commonly known as muscle contraction to cross-link actin, myosin, and muscle fiber [12]. In addition to muscle contraction, Ca2+ signaling can regulate the neuronal system, immune system, and fluid secretion [13-15]. In addition, Ca2+ signaling is critical for physiological and pathological cellular pathways, including cell proliferation, differentiation, and migration”. Please rewrite these sentences.
Pag. 3: “RyR is activated by ryanodine”…This is only partially correct. Ryanodine is the natural activator, but other drugs/molecules are able to activate these receptors.
Chapter 2: “LGCC are found with numerous types and functional roles [29], including ATP-stimulated P2X channels [30], ionotropic glutamate receptors (e.g., α-amino-3-hy-droxy-5-methyl-4-isoxazolepropionic acid receptors or N-methyl-D-aspartate receptors) that induce synaptic transmission [31,32]”. Please rewrite these sentences.
The whole manuscript needs a careful revision
Author Response
Dear reviewer and editor,
Before addressing each of the comments below, we appreciate the reviewers for the valuable comments and careful consideration. We obviously have needed to quote all sources correctly and done so at the places where we had missed before. In addition, the manuscript has been edited to make appropriate information and additional data to this body of work.
Responses to comments of reviewer as below:
Reviewer 2
This review paper summarizes the basic mechanisms of calcium selective channels implicated in breast cancer and their role in metastatic stages of this malignance.
The topic touched upon in the article is relevant. The scientific content of the manuscript could justify its publication, but this manuscript requires thorough revision with extensive rewriting (possibly availing to the English editing service).
This paper would benefit from some closer proofreading.
It includes many linguistic errors (e.g. agreement of verbs) that at times make it difficult to follow.
It may be useful to engage a professional English language editor following a restructure of the paper.
For instance:
Simple summary: ….”blockers are proposed against breast cancers and provide”….should be ….”blockers have been proposed against breast cancers and provide”
-Response: We appreciate your valuable comment and the simple summary was edited as you recommended.
Abstract: The last sentences starting with “Thus, we demonstrated….are hard to follow. Please rewrite it in a clearer way.
-Response: We appreciate your valuable comment and the abstract was edited as you recommended.
Introduction, line 4: “that is transformed into a mesenchymal cell-like morphology owing to EMT”. This sentence is hard to follow. Please rephrase.
-Response: We appreciate your valuable comment and the abstract was edited as you recommended (line 38).
Introduction: “In this review, we elucidate the essential process of metastasis in breast cancer. In particular, Ca2+ signaling molecules are introduced and Ca2+ signaling-associated breast cancer metastasis is focused upon. This review also describes the fundamental modulation of Ca2+ signaling modules and discusses potential strategies against breast cancer”. Please rewrite these sentences-
-Response: We appreciate your valuable comment and the abstract was edited as you recommended (line 55).
Chapter 1.1: “The physiological role of Ca2+ signaling is commonly known as muscle contraction to cross-link actin, myosin, and muscle fiber [12]. In addition to muscle contraction, Ca2+ signaling can regulate the neuronal system, immune system, and fluid secretion [13-15]. In addition, Ca2+ signaling is critical for physiological and pathological cellular pathways, including cell proliferation, differentiation, and migration”. Please rewrite these sentences.
-Response: We appreciate your valuable comment and these sentences (line 60) was edited as you recommended.
Pag. 3: “RyR is activated by ryanodine”…This is only partially correct. Ryanodine is the natural activator, but other drugs/molecules are able to activate these receptors.
-Response: We appreciate your valuable comment and edited this part and added activators in this section (line 75).
Chapter 2: “LGCC are found with numerous types and functional roles [29], including ATP-stimulated P2X channels [30], ionotropic glutamate receptors (e.g., α-amino-3-hy-droxy-5-methyl-4-isoxazolepropionic acid receptors or N-methyl-D-aspartate receptors) that induce synaptic transmission [31,32]”. Please rewrite these sentences.
-Response: We appreciate your valuable comment and these sentences (line 85) was edited as you recommended.
The whole manuscript needs a careful revision
-Response: We appreciate your valuable comment and we rephrased the sentences in whole manuscript, represented detailed type of breast cancer as you recommended.
Reviewer 3 Report
The paper entitled 'Ca2+ Signaling as “Untact Mode” during Signaling in Metastatic Breast Cancer' by Lee and Hong is a review that adresses an interesting topic.
The simple summary is difficult to follow by a reader that is not familiar with the abbreviations (e.g. SOCC, PMCA, SPCA, EMT etc.).
The information indicated by the authors is sometimes misleading and several times it was used the term 'Ca2+ channels' to identify other molecular components, such as receptors, pumps etc. Some examples to argument this idea
- The section '1.2. Types of Ca2+ channels in the plasma membrane' contains several Ca2+ signaling pathways and is not limited to Ca2+ channels.
- In the section '2. The relationship between breast cancer metastasis and Ca2+ channels' is the same problem. Additionnally, the sentence 'Numerous Ca2+ channels are overexpressed in breast cancer cells, including IP3R [50], Orais [51-53], PMCA [54,55], and TRP channels [56-59].' is confusing. IP3R and PMCA are not calcium channels.
I also recommend the authors to read the book chapter by Dumitru et al. that extensively reviews the Alterations in Calcium Signaling Pathways in Breast Cancer, Calcium and Signal Transduction, DOI: 10.5772/intechopen.80811. (Available from: https://www.intechopen.com/books/calcium-and-signal-transduction/alterations-in-calcium-signaling-pathways-in-breast-cancer) and use the necessary information to improve their paper.
Author Response
Dear reviewer and editor,
Before addressing each of the comments below, we appreciate the reviewers for the valuable comments and careful consideration. We obviously have needed to quote all sources correctly and done so at the places where we had missed before. In addition, the manuscript has been edited to make appropriate information and additional data to this body of work.
Responses to comments of reviewer as below:
Reviewer 3
The paper entitled 'Ca2+ Signaling as “Untact Mode” during Signaling in Metastatic Breast Cancer' by Lee and Hong is a review that adresses an interesting topic.
The simple summary is difficult to follow by a reader that is not familiar with the abbreviations (e.g. SOCC, PMCA, SPCA, EMT etc.).
-Response: We appreciate your valuable comment and the simple summary was edited as you recommended.
The information indicated by the authors is sometimes misleading and several times it was used the term 'Ca2+ channels' to identify other molecular components, such as receptors, pumps etc. Some examples to argument this idea
- The section '1.2. Types of Ca2+ channels in the plasma membrane' contains several Ca2+ signaling pathways and is not limited to Ca2+ channels.
-Response: We appreciate your valuable comment and subtitle (line 85) was rephrased to Types of Ca2+ channels.
- In the section '2. The relationship between breast cancer metastasis and Ca2+ channels' is the same problem. Additionnally, the sentence 'Numerous Ca2+ channels are overexpressed in breast cancer cells, including IP3R [50], Orais [51-53], PMCA [54,55], and TRP channels [56-59].' is confusing. IP3R and PMCA are not calcium channels.
-Response: We appreciate your valuable comment and apologized the confusion. We edited line 120 to represent the clear description.
I also recommend the authors to read the book chapter by Dumitru et al. that extensively reviews the Alterations in Calcium Signaling Pathways in Breast Cancer, Calcium and Signal Transduction, DOI: 10.5772/intechopen.80811. (Available from: https://www.intechopen.com/books/calcium-and-signal-transduction/alterations-in-calcium-signaling-pathways-in-breast-cancer) and use the necessary information to improve their paper.
-Response: We appreciate your valuable comment and suggestion. The book chapter is helpful to revise our manuscript and rephrased several points in the text. In this review, we focused on the characteristics of breast cancer with a view of Ca2+ signaling and cancer metastasis, especially from breast to brain, and discussed potential strategies to overcome the disadvantages of breast cancer-targeted therapy with a view of Ca2+ signaling.
Round 2
Reviewer 1 Report
This new version is really better :)
I have only few minor concerns now:
- line 50-51.the authors should precise what means triple negative and triple-positive vs hormone receptor-positive
- line 86: the authors probably mean P2Y channles instead P2X channels as it s written ?
- line 149-150: is there any explanation why ATP does not stimulate P2Y2R of MCF-7 cells ?
Author Response
Dear reviewer and editor,
Before addressing each of the comments below, we appreciate the the valuable comments for second revision. The manuscript has been edited to make appropriate information to this body of work.
Responses to comments of reviewer as below:
- line 50-51.the authors should precise what means triple negative and triple-positive vs hormone receptor-positive
Response: We appreciate your valuable comment. We revised the manuscript and added the meaning of hormone receptor-positive cell in line 50 as follows; triple-negative (MDA-MB-231, MDA-MB-486, and MCF-10A [16,17]), triple-positive (BSMZ, BT474, and EFM192A [16]), and hormone receptor-positive cell lines which express estrogen receptors and progesterone receptors in the absence of HER2 (MCF-7 and T47D [16]). Genotypic or phenotypic heterogeneity of breast cancer is diverse. While triple-negative breast cancer generally has the most aggressive behavior and poor clinical outcomes [18-20], triple-positive breast cancer has also been found to exhibit aggressive behavior, despite the availability of antibody-targeted therapy or chemotherapy [21].
- line 86: the authors probably mean P2Y channels instead P2X channels as it’s written ?
Response: We appreciate your valuable comment. LGCC are composed of various receptors such as ATP receptors, GABA, and 5-HT receptors. P2Y(GPCR) channels as well as P2X channels are also included in line 88. Thus, we rephrased the ATP receptors instead of P2X channels.
- line 149-150: is there any explanation why ATP does not stimulate P2Y2R of MCF-7 cells ?
Response: We appreciate your valuable comment. We added the reason in line 153 why P2Y2 isn’t stimulated in MCF-7 as follows; MCF-7 (estrogen receptor-positive) cells, which has low metastatic feature, induces less release of ATP and reveals low P2Y2 receptor activation.
Reviewer 2 Report
The authors have adequately addressed the reviewer points of critique in their revised manuscript. I recommend the publication of this manuscript in its current form
Author Response
We appreciate your favorable consideration.
Reviewer 3 Report
The manuscript was revised in accordance with the suggestions. I consider that it can be published in the present form.
Author Response

(The authors gave the same response as above.)
